# The Maize Gene *ZmGLYI-8* Confers Salt and Drought Tolerance in Transgenic *Arabidopsis* Plants

**DOI:** 10.3390/ijms252010937

**Published:** 2024-10-11

**Authors:** Ting Yu, Wei Dong, Xinwei Hou, Aiqing Sun, Xinzheng Li, Shaowei Yu, Jiedao Zhang

**Affiliations:** 1State Key Laboratory of Crop Biology, College of Life Sciences, Shandong Agricultural University, Tai’an 271018, China; 15254887381@163.com (T.Y.); 15269706237@163.com (W.D.); lxz@sdau.edu.cn (X.L.); 2Maize Research Institute, Shandong Academy of Agricultural Sciences, Jinan 250100, China; 15953852159@163.com; 3State Key Laboratory of Crop Biology, College of Agronomy, Shandong Agricultural University, Tai’an 271018, China; saqsshh@sdau.edu.cn; 4State Key Laboratory of Wheat Improvement, College of Life Sciences, Shandong Agricultural University, Tai’an 271018, China

**Keywords:** glyoxalase I, methylglyoxal, salt tolerance, drought tolerance, maize

## Abstract

Methylglyoxal (MG), a highly reactive and cytotoxic α-oxoaldehyde compound, can over-accumulate under abiotic stress, consequently injuring plants or even causing death. Glyoxalase I (GLYI), the first enzyme of the glyoxalase pathway, plays multiple roles in the detoxification of MG and in abiotic stress responses. However, the *GLY1* gene in maize has been little studied in response to abiotic stress. In this study, we screened a glyoxalase I gene (*ZmGLYI-8*) and overexpressed in *Arabidopsis*. This gene was localized in the cytoplasm and can be induced in maize seedlings under multiple stress treatments, including salt, drought, MG, ABA, H_2_O_2_ and high temperature stress. Phenotypic analysis revealed that after MG, salt and drought stress treatments, overexpression of *ZmGLYI-8* increased the tolerance of transgenic *Arabidopsis* to MG, salt and drought stress. Furthermore, we demonstrated that the overexpression of *ZmGLYI-8* scavenges accumulated reactive oxygen species, detoxifies MG and enhances the activity of antioxidant enzymes to improve the resistance of transgenic *Arabidopsis* plants to salt and drought stress. In summary, this study preliminarily elucidates the molecular mechanism of the maize *ZmGLYI-8* gene in transgenic *Arabidopsis* and provides new insight into the breeding of salt- and drought-tolerant maize varieties.

## 1. Introduction

With the threat of global climate change, the frequency and severity of abiotic stresses such as low temperature, high temperature, drought and salinity in plants are gradually increasing. These abiotic stresses disrupt physiological and metabolic balance and consequently interfere with plant growth and development [1]. Approximately 10% of the world’s arable land is affected by abiotic stresses, resulting in yield losses of more than 50% for important crops such as wheat, maize and rice [2]. To combat the deleterious effects of abiotic stress, plants adapt to adverse conditions mainly through physiological and biochemical processes such as ion homeostasis, osmotic adjustment, reactive oxygen species (ROS) scavenging and nutritional balance [1].

The accumulation of toxic metabolites is a significant cause of abiotic stress damage to plants. When plants are exposed to abiotic stress, the cellular homeostasis of ROS and MG in plants is inevitably disturbed [3,4]. As a primary consequence of abiotic stresses, ROS can be overproduced by impairing stomatal conductance, increasing photorespiration and inactivating photosynthesis enzymes and pigments [5,6]. High concentrations of ROS cause plasma membrane peroxidation, protein degradation and DNA mutation, ultimately leading to cell death [3,7,8]. MG, a toxic α-oxoaldehyde, is mainly generated as a byproduct of the glycolytic pathway, pentose phosphate pathway and Calvin–Benson cycle [4,9,10]. Under abiotic stresses of low temperature, salt, drought and heavy metals, MG levels increased two- to six-fold in the seedlings of rice, *Pennisetum alopecuroides*, tobacco, *oilseed rape* and *pumpkin* [11]. Overaccumulation of MG inactivates the antioxidant system and results in the formation of advanced glycation end products (AGEs) with nucleotide and protein moieties [4,12,13,14,15].

Plants can scavenge toxic metabolites either through enzymatic or nonenzymatic pathways. ROS can be detoxified by nonenzymatic antioxidants, such as ascorbate (ASH), glutathione (GSH) and carotenoids, or by antioxidant enzymes, such as superoxide dismutase (SOD), catalase (CAT), guaiacol peroxidase (GPX) and ascorbate peroxidase (APX) [16,17,18].

The glyoxalase system, which is composed of glyoxalase I (GLYI), glyoxalase II (GLYII) and glyoxalase III (GLYIII), participates in the detoxification of MG [11,19,20,21]. In the presence of GSH, over-accumulated MG is converted by GLYI to S-D-lactoylglutathione (S-LG), which is then hydrolyzed to the nontoxic compound D-lactate by GLYII [22]. GlyIII can also convert MG into D-lactate through a single step [23].

Studies have shown that the GLYI-GLYII pathway may play a major role in the scavenging of MG and resistance to abiotic stresses [10,11,23,24]. Multiple glyoxalase genes, such as those involved in salinity, osmotic and oxidative stress, have been shown to confer tolerance to various abiotic stresses in crop plants [20,25,26]. Overexpression of the *B. juncea* GLYI gene (*BjGLYI)* significantly enhanced the tolerance of tobacco to high salt concentrations and excessive MG [27].The loss of *AtGLYI-2* in *Arabidopsis* leads to sensitivity to MG and salt stress [28]. The overexpression of the *Brassica juncea GLYI* gene in rice improved the salt tolerance of the transgenic plants [29]. Similarly, the *GlyI* gene was cloned and characterized from *Brassica juncea*, and its expression was upregulated under salt and water stress [30]. *GLYI* transcripts were significantly upregulated in tomato roots, stems and leaves under ABA and mannitol treatment [31]. *GLYI* transcripts was induced by salinity, MG, white light and heavy metal treatments in pumpkin seedlings [32]. In wheat, the expression levels of *GLYI* were upregulated after NaCl and ZnCl_2_ treatments [33]. Analysis of salt-tolerance traits in tolerant and sensitive varieties of tomato and rice revealed that the stress-induced *GLYI* gene had a high expression level in tolerant varieties [34]. Rice tolerance to salinity stress and excess MG could be enhanced by overexpressing the *GLYII* gene [29]. When *Brassica juncea* plants were exposed to ABA, ZnCl_2_ and NaCl, the expression level of *GLYII* increased three- to four-fold [35]. At present, although a few studies have confirmed the ability of multiple glyoxalase genes to enhance abiotic stress resistance in plants, the underlying genetic mechanism is unclear. Whether the remaining 13 *GLYI* genes and 3 *GLYII* genes in maize respond to abiotic stress has not been reported.

In this study, we aimed to investigate the functions of the *ZmGLYI-8* gene in response to salt and drought stress. We cloned the *ZmGLYI-8* gene from the maize inbred line B73. *ZmGLYI-8* gene was induced by salt, drought, MG, ABA, H_2_O_2_ and high-temperature stress. Overexpression of *ZmGLYI-8* in *Arabidopsis* significantly improved the tolerance of transgenic plants to salt stress, and the seedlings had longer root length, lower wilting degree and higher chlorophyll content. In addition, the activities of CAT, POD and SOD in transgenic plants also increased, but the contents of MG, MDA, O_2_^•−^ and H_2_O_2_ decreased under salt and drought stress. Overall, we demonstrated that overexpression of the *ZmGLYI-8* gene enhanced tolerance to salt and drought stress in transgenic *Arabidopsis* plants by detoxifying MG, increasing antioxidant activity and reducing ROS production. These results provide novel insight into the function of the *GLYI* gene under abiotic stress in maize.

## 2. Results

### 2.1. Cloning and Sequence Analysis of ZmGLYI-8

In this study, the glyoxalase gene *ZmGLYI-8* (LOC100281531) was cloned from the maize inbred line B73. The complete coding sequence of *ZmGLYI-8* was 1044 bp, encoding 347 amino acids with a 38.22-kDa molecular mass and 6.49 isoelectric point (Appendix A). The unique lactoylglutathione lyase domains (PF00903) of glyoxalase I were identified at amino acid sequences 84–205 and 217–335 of *ZmGLYI-8* by using NCBI CD-Search. Based on the amino acid sequence of *ZmGLYI-8*, homologous sequences in other species, including *Arabidopsis* (NP-176896.1), *Brachypodium distachyon* (XM-003508671.1), *Glycine max* (XP-003530499), *Hordeum vulgare* (AK364964.1), *Oryza sativa* (LOC4338161) and *Sorghum bicolor* (XM002440785.1), were screened by using the NR database of the BLAST program (Figure 1a), and the sequence similarity was 82.84%. Comparative analysis of multiple sequences revealed that the amino acid sequences of *ZmGLYI-8* and its homologs were highly conserved in the middle and C-terminus of the sequence. Phylogenetic analysis revealed that *ZmGLYI-8* was closely related to *OsGLYI* of *Oryza sativa* (Figure 1b). The evolutionary relationship of *ZmGLYI-8* and its homologs is consistent with the process of plant evolution in nature.

### 2.2. ZmGLYI-8 Responds to a Variety of Abiotic Stresses and Hormones

To investigate the potential function of *ZmGLYI-8*, the expression pattern of *ZmGLYI-8* in the roots, stems and leaves of B73 at the V6 stage was measured via qRT–PCR (Figure 2g). *ZmGLYI-8* was expressed in all three tissues. The expression of *ZmGLYI-8* in the leaves and stems was 51 and 32 times greater than that in the roots, respectively. These results showed that *ZmGLYI-8* may perform essential functions in the leaves and stems of B73.

In addition, the expression patterns of *ZmGLYI-8* in the leaves of B73 under multiple abiotic stresses and hormones were also investigated (Figure 2a–h). After 150 mM NaCl, 10% PEG-6000 and HT treatments, the expression levels of *ZmGLYI-8* peaked at 6 h and were 2.2, 2.6 and 1.7 times greater than those in the control, respectively. Then, the expression level of *ZmGLYI-8* gradually decreased from 6 to 24 h. Under 10 mM MG treatment, the expression level of *ZmGLYI-8* peaked at 1 h (approximately 1.4 times higher than that at 0 h), sustained high expression from 1 to 12 h and then significantly decreased at 24 h. Under 50 µM ABA and 0.5 mM H_2_O_2_ treatments, the expression level of *ZmGLYI-8* peaked at 1 h (approximately 2.9 and 1.7 times higher than that at 0 h) and then decreased from 1 to 24 h. After 5 mM ET treatment, the expression level of *ZmGLYI-8* did not obviously change before 1 h and decreased significantly from 1 to 24 h. After 50 µM MeJA treatment, the expression level of *ZmGLYI-8* did not change significantly from 1 to 24 h. These results indicated that *ZmGLYI-8* plays an important role in the response to multiple abiotic stresses.

### 2.3. ZmGLYI-8 Localizes to the Cytoplasm

According to the analysis results of the SubLoc program, the *ZmGLYI-8* protein was predicted to be localized in the cytoplasm. To further validate the subcellular localization of *ZmGLYI-8*, 35S::*ZmGLYI-8*-GFP or 35S::GFP plasmids were expressed instantaneously in *Nicotiana benthamiana* leaves. The GFP signal of 35S::GFP was detected in the cytoplasm and nucleus, whereas the GFP signal of 35S::*ZmGLYI-8*-GFP was observed only in the cytoplasm (Figure 3). Therefore, the *ZmGLYI-8* protein functions in the cytoplasm. This result was consistent with the prediction of the SubLoc (http://www.bioinfo.tsinghua.edu.cn/SubLoc/ (accessed on 5 October 2014)) program.

### 2.4. ZmGLYI-8 Enhances Tolerance to MG, Salt and Drought Stresses in Prokaryotes

To investigate the possible function of *ZmGLYI-8*, the pET-30a vector with the full-length CDS of *ZmGLYI-8* was transformed into *Escherichia coli* Rosetta. Double-digestion experiments with both restriction enzymes showed that the recombinant prokaryotic expression vector pET-30a-*ZmGLYI-8* was constructed successfully. Sodium dodecyl sulfate–polyacrylamide gel electrophoresis (SDS–PAGE) was used to detect the *ZmGLYI-8* protein in the Rosetta strain containing pET-30a-*ZmGLYI-8* (Appendix A). Different concentrations of IPTG-induced bacteria were added dropwise to LB solid medium supplemented with 0.5 mM MG, 200 mM NaCl or 200 mM mannitol. As shown in Appendix A, compared with the control strain pET-30a, the recombinant strain pET-30a-*ZmGLYI-8* grew better on LB solid medium supplemented with 0.5 mM MG, 200 mM NaCl or 200 mM mannitol. These results indicate that the *ZmGLYI-8* gene enhances tolerance to MG, salt and drought stresses in prokaryotes.

### 2.5. Overexpression of ZmGLYI-8 in Arabidopsis Confers Tolerance to MG Stress

To further verify whether *ZmGLYI-8* responded to abiotic stress in plants, *ZmGLYI-8* with CaMV 35S was transformed into *Arabidopsis*. The relative expression level of *ZmGLYI-8* in transgenic *Arabidopsis* plants was analyzed via RT–PCR. Three highly expressed transgenic lines (OE3, OE4 and OE5) were screened from T_3_ homozygous lines for subsequent experimental analysis (Appendix A). The expression level of *ZmGLYI-8* in OE5 transgenic *Arabidopsis* plants was the highest among all the transgenic lines. To investigate the tolerance of *ZmGLYI-8* transgenic lines to MG, different concentrations of MG were applied to 1/2 MS medium to measure the germination rates and root lengths of the *ZmGLYI-8* transgenic lines (Figure 4). After 7 days of treatment, there was no significant difference in the seed germination rates of OE3, OE4 and OE5 compared to those of the WT on 1/2 MS medium without MG. However, the seed germination of both the WT and *ZmGLYI-8* transgenic plants was inhibited on 1/2 MS medium supplemented with 2.5 mM MG. The seed germination rates of the OE3, OE4 and OE5 transgenic lines were 97.5, 94 and 98%, respectively. However, the germination of the WT was severely suppressed. Compared to those of OE3, OE4 and OE5, the seed germination rate of the WT was only 88%. On 1/2 MS medium without MG, the lengths of the roots of the OE3, OE4 and OE5 plants were not significantly different from those of the WT plants. Treatment with 0.5 mM MG led to reduced root lengths in both the WT and *ZmGLYI-8* transgenic plants. The root lengths of OE3, OE4 and OE5 were 12.40, 11.31 and 12.58 mm, respectively. The degree of suppression of WT root length (7.98 mm) was greater than that of OE3, OE4 and OE5. These results showed that overexpression of *ZmGLYI-8* enhanced the tolerance of transgenic *Arabidopsis* plants to MG.

### 2.6. Overexpression of ZmGLYI-8 in Arabidopsis Confers Tolerance to Salt Stress

To investigate the tolerance of *ZmGLYI-8* transgenic lines to salt stress, five-day-old seedlings were vertically cultivated on 1/2 MS solid medium with or without 150 mM NaCl for salt sensitivity analysis (Figure 5). The root lengths of the WT and three transgenic *Arabidopsis* lines did not significantly differ on 1/2 MS solid medium without 150 mM NaCl. However, on 1/2 MS medium supplemented with 150 mM NaCl, the roots of OE3, OE4 and OE5 were significantly longer than those of the WT. The roots of OE3, OE4 and OE5 (33.23 ± 0.34, 29.91 ± 6.74 and 35.18 ± 2.37 mm, respectively) were ~3 times longer than those of the WT (11.30 ± 1.85 mm) (Figure 5a,b). Moreover, the salt tolerance of adult transgenic *Arabidopsis* seedlings was also verified. There were no significant differences in growth between the WT and transgenic *Arabidopsis* lines under normal conditions. However, under the 400 mM NaCl treatment, OE3, OE4 and OE5 grew better than the WT, and the degree of leaf wilt of OE3, OE4 and OE5 was lower than that of the WT (Figure 5c). In addition, leaf discs (1 cm in diameter) were placed in water (simulated control) or under salt stress conditions (simulated stress treatment), including salt stress at 200 mM and 400 mM NaCl, to investigate the tolerance of *ZmGLYI-8* transgenic lines to salt stress. As shown in Figure 6d, the leaf discs of the OE3, OE4, OE5 and WT plants remained green under control conditions. Under salt stress, the leaf discs of the OE3, OE4, OE5 and WT plants turned yellow, but the degree of leaf disc yellowing in the OE3, OE4 and OE5 plants was less than that in the WT plants. Although the chlorophyll content in the leaf discs of both the WT and transgenic *Arabidopsis* lines decreased after salt stress treatment, the decrease in the chlorophyll content in the leaf discs of the WT plants was significantly greater than that in the leaf discs of the OE3, OE4 and OE5 plants (Figure 5e). These results indicated that the salt tolerance of the transgenic *Arabidopsis* lines was greater than that of the WT plants, indicating that overexpression of *ZmGLYI-8* enhanced the salt tolerance of *Arabidopsis* plants.

To further explore how *ZmGLYI-8* enhances the salt tolerance of transgenic *Arabidopsis* plants, the contents of O_2_^•−^ and H_2_O_2_ in transgenic *Arabidopsis* lines under salt stress treatment were measured. The levels of O_2_^•−^ and H_2_O_2_ were lower in the transgenic *Arabidopsis* lines than in the WT plants under salt stress treatment (Figure 6a,b), and the levels of MDA in the transgenic *Arabidopsis* lines were also significantly lower than those in the WT plants. These results showed that the overexpression of *ZmGLYI-8* alleviated salt stress damage by detoxifying ROS in *Arabidopsis*. Additionally, the MG content of both the WT and transgenic *Arabidopsis* lines was measured. There was no significant difference in the MG content between the WT and transgenic *Arabidopsis* lines in the control treatment group. However, the MG content was significantly lower in OE3, OE4 and OE5 (58.17 ± 2.10, 68.57 ± 1.07 and 52.81 ± 1.30 μg/gFW, respectively) than in the WT (72.48 ± 2.15 μg/gFW) (Figure 6f). These results showed that overexpression of *ZmGLYI-8* enhanced the salt stress tolerance of *Arabidopsis* plants by alleviating the over-enrichment of MG. Furthermore, we determined the activity of three antioxidant enzymes (CAT, POD and SOD) under salt stress conditions. The activity of these three antioxidant enzymes in the transgenic *Arabidopsis* lines was greater than that in the WT plants under salt stress treatment, whereas the activity of these enzymes in both the WT and transgenic *Arabidopsis* lines showed no difference under the control treatment (Figure 6c–e). Among them, the activity of antioxidant enzymes in OE5 was highest among the three transgenic *Arabidopsis* lines. These results suggested that the transgenic *Arabidopsis* lines possessed greater tolerance to salt stress than did the WT because of MG detoxification and increased antioxidant enzyme activity.

### 2.7. Overexpression of ZmGLYI-8 in Arabidopsis Confers Tolerance to Drought Stress

To verify whether *ZmGLYI-8* responded to drought stress, the root lengths of the WT and three transgenic *Arabidopsis* lines (OE3, OE4 and OE5) were measured on 1/2 vertical MS solid medium supplemented with 150 mM mannitol, and the results showed that the root lengths of OE3, OE4 and OE5 (50.41 ± 2.60, 47.27 ± 2.19 and 55.08 ± 2.83 mm, respectively) were significantly greater than that of the WT (33.25 ± 4.64 mm). There were no significant differences in root length between the WT and transgenic *Arabidopsis* lines under the control treatment (Figure 7a,b). Moreover, under the 400 mM mannitol treatment, the growth of adult OE3, OE4 and OE5 plants was greater than that of the WT plants, and OE3, OE4 and OE5 plants exhibited less leaf wilt than did the WT plants (Figure 7c). In addition, the degree of leaf disc yellowing in the OE3, OE4 and OE5 plants was less than that in the WT plants, and the chlorophyll content in the leaf discs of the OE3, OE4 and OE5 plants was significantly greater than that in the leaf discs of the WT plants under the 200 mM and 400 mM mannitol treatments (Figure 7d,e). These results showed that overexpression of *ZmGLYI-8* conferred greater tolerance to drought stress in *Arabidopsis*.

To further investigate the drought tolerance regulatory mechanisms of *ZmGLYI-8*, the contents of O_2_^•−^, H_2_O_2_ and MDA in the OE3, OE4 and OE5 lines were measured under drought stress conditions. The contents of O_2_^•−^ and H_2_O_2_ were significantly greater in the WT than in the OE3, OE4 and OE5 plants. Moreover, the MDA and MG contents in the leaves of OE3, OE4 and OE5 plants were significantly lower than those in the leaves of WT plants under drought stress (Figure 8f). Furthermore, there was greater activity of CAT, POD and SOD in the OE3, OE4 and OE5 plants than in the WT plants under drought stress (Figure 8e,f). These results showed that *ZmGLYI-8* enhanced the drought resistance of *Arabidopsis* plants by decreasing the ROS and MG contents and increasing the activity of antioxidant enzymes.

## 3. Discussion

MG, a cytotoxic and mutagenic compound, is a byproduct of various metabolic activities in organisms. Under abiotic stress, the MG concentration rapidly increased two- to six-fold, disrupting normal cellular function, altering metabolic behavior and even leading to plant death [11,36]. To maintain the dynamic balance of MG in plant cells, the glyoxalase system, a unique and effective MG detoxification system, developed during plant evolution [37]. The transcription and protein expression of GLYI are induced in plants in response to various abiotic stresses. Salt and drought stress are the main environmental stress factors limiting crop yield and can lead to MG stress, oxidative stress and osmotic stress. Increasing the activities of glyoxalase and antioxidant enzymes and accumulating osmoregulatory substances are important strategies for plant resistance to salt and drought stress [28,30,32,38]. We cloned the *ZmGLYI-8* gene from B73, which has a highly conserved amino acid sequence and two unique lactoylglutathione lyase domains (Figure 1). The ZmGLYI-8 protein was localized in the cytoplasm (Figure 3). We also demonstrated that the *ZmGLYI-8* gene responds to multiple abiotic stresses (Figure 2).

The glyoxalase pathway is primarily associated with abiotic stress tolerance mechanisms in plants. The overexpression of glyoxalase pathway genes in plants has been shown to increase plant tolerance to salt, drought, MG, high temperature, low temperature, heavy metals and MG stress [22,27,30,39,40,41]. We found that the *ZmGLYI-8* gene enhances tolerance to MG, salt and drought stresses in prokaryotes (Appendix A) and speculated that the *ZmGLYI-8* gene is closely associated with adaptation to salt and drought stress in eukaryotes. Then, we cloned the *ZmGLYI-8* coding sequence from B73 and subsequently obtained transgenic *Arabidopsis* plants with *ZmGLYI-8* to investigate the molecular mechanisms of salt and drought tolerance in transgenic *Arabidopsis* plants. In this study, the seed germination rates and root lengths of the transgenic *Arabidopsis* lines were significantly greater than those of the WT plants under the MG treatment (Figure 4b,d). The results showed that overexpression of *ZmGLYI-8* enhanced tolerance to MG in transgenic *Arabidopsis* plants. In addition, we also found that overexpression of the *ZmGLYI-8* gene enhanced the salt and drought tolerance of transgenic *Arabidopsis* plants. Compared with the WT plants, the OE3, OE4 and OE5 plants had longer roots, greater growth, lower degrees of leaf chlorosis and greater chlorophyll contents under salt and drought stress (Figure 5 and Figure 7).

Both ROS and MG are phytotoxic to plants under abiotic stresses due to the induction of oxidative stress. The production of MG is enhanced under abiotic stress, leading to excessive generation of ROS in plant cells. Moreover, MG promotes ROS formation through the formation of advanced glycation end products (AGEs). Consequently, elevated levels of MG contribute significantly to the development of oxidative stress [42,43,44]. Therefore, elimination of MG can effectively inhibit ROS production induced by MG [10]. The antioxidant and glyoxalase systems are involved in ROS detoxification, so the coordination of these systems can reduce oxidative stress by detoxifying both ROS and MG, respectively [10]. Many studies have shown that the antioxidant defense system and the glyoxalase system mitigate oxidative stress by detoxifying ROS and MG, respectively. Under abiotic stress, exogenously applied phytoprotectants have been shown to enhance tolerance in *Oryza sativa* [45,46,47], *Triticum aestivum* [48,49] and *Brassica napus* [50,51] seedlings through the coordinated action of the upregulated antioxidant defense system and glyoxalase system. ROS homeostasis is very important for plant growth and development. Abiotic stress induces cell overproduction of ROS, leading to plant cell damage; therefore, plants need to precisely control ROS production when necessary [8]. Enzymatic and nonenzymatic antioxidant defense systems remove reactive oxygen species and protect plant cells from oxidative damage [3,38]. To further elucidate the molecular mechanism of the *ZmGLYI-8* gene response to salt and drought stress, the contents of O_2_^•−^, H_2_O_2_ and MDA in the transgenic *Arabidopsis* lines under both salt and drought stress treatments were measured. We found that the O_2_^•−^, H_2_O_2_ and MDA levels in the transgenic *Arabidopsis* lines were lower than those in the WT plants (Figure 6a,b and Figure 8a,b). CAT, SOD and POD are enzymatic antioxidant systems that regulate ROS homeostasis in plants and participate in the reduction of O_2_^•−^ to H_2_O_2_ [52,53]. In our study, CAT, SOD and POD levels were significantly greater in the transgenic *Arabidopsis* lines than in the WT plants under both salt (Figure 6c–e) and drought (Figure 8c–e) treatments. The findings suggest that *ZmGLYI-8* enhances plant tolerance to salt and drought stress through the detoxification of MG, augmentation of antioxidant enzyme activity and reduction in ROS production.

When plants are subjected to abiotic stress, MG levels are elevated [11,54]. ROS are highly reactive and MG is a potent reactive cytotoxic capable of completely disrupting cellular functions, including lipid peroxidation, protein oxidation, fatty acid oxidation and disruption of biofilm structure and functions [13,52]. MG overproduction inhibits germination and cell proliferation and leads to protein glycation, dysregulation of antioxidant defenses and other metabolic dysfunctions [15,55,56,57]. The glyoxalase pathway enhances plant resistance to abiotic stress by detoxifying MG [27,58]. Co-expression of the *SlGLYI4* and *SlGLYII2* genes in tomato enhanced the tolerance of transgenic plants to salt and osmotic stress by directly inhibiting the accumulation of MG [24]. In *mung bean* seedlings, proline and glycinebetaine can activate the glyoxalase systems GLYI and GLYII, thereby reducing plant MG levels and improving salt tolerance [59]. In rice seedlings, the salt-tolerant cultivar (Pokkali) had greater GLYI and GLYII activities and lower MG contents than did the salt-sensitive cultivar (IR64) under NaCl stress [60]. In our study, the MG content was significantly lower in the transgenic *Arabidopsis* lines than in the WT plants (Figure 6f and Figure 8f). These results indicate that overexpression of the *ZmGLYI-8* gene enhances resistance to salt and drought stress in the transgenic *Arabidopsis* plants, mainly through the coordinated actions of antioxidant defense and glyoxalase systems in mitigating oxidative stress by detoxifying both ROS and MG, respectively.

Maize is one of the most important food and feed crops and has important economic significance. With the intensification of global climate change, crops inevitably suffer from various abiotic stresses, such as salt, drought and high temperatures, during their growth and development, which seriously affects crop yield and quality [61,62]. When plants are subjected to abiotic stress, intracellular MG and ROS are overproduced, threatening the growth and development of plants. Both antioxidant and glyoxalase systems play important roles in the detoxification of ROS, and thus coordination of these systems can reduce ROS production, thereby alleviating oxidative stress and increasing plant tolerance to abiotic stresses [47,63,64]. In this study, we demonstrated the role of *ZmGLYI-8* gene in regulating abiotic stress tolerance in *Arabidopsis* by analyzing expression, phenotypic, physiological and biochemical data, in which overexpression of the *ZmGLYI-8* gene reduces ROS production by detoxifying MG and increasing antioxidant enzyme activity, thereby enhancing the tolerance of transgenic *Arabidopsis* to salt and drought stress (Figure 9). Our study provided the theoretical basis for the application of *GLYI* gene in genetic improvement of crops for sustainable agricultural development. In future work, we will verify the function of the *ZmGLYI-8* gene in maize or other crops and hope to widely apply the *ZmGLYI-8* gene in the breeding of salt- and drought-tolerant crop varieties.

## 4. Materials and Methods

### 4.1. Gene Isolation and Bioinformatics Analysis of ZmGLYI-8

Based on the B73 genome in the MaizeGDB database, Primer 5.0 software was used to design specific primers for *ZmGLYI-8* (Appendix A). Total RNA was extracted from the leaves of ten-day-old maize seedlings with TRIzol reagent. The concentration and integrity of total RNA were determined by using formaldehyde gel electrophoresis and a NanoDrop 2000 spectrophotometer (Thermo Scientifi, Shanghai, China). CDNA from B73 leaves was synthesized by using HiScript^®^II Q Select RT SuperMix (Vazyme, Nanjing, China). The pMD19-T cloning vectors with the PCR products of *ZmGLYI-8* were transformed into *Escherichia coli* DH5α, which was subsequently sequenced by Boshang (Shanghai, China). The molecular weight and isoelectric point of *ZmGLYI-8* were evaluated by ProtParam (http://web.expasy.org/protparam/ (accessed on 1 October 2024)), and the conserved functional domains of this gene were predicted by using NCBI CD-Search (https://www.ncbi.nlm.nih.gov/cdd/ (accessed on 1 October 2024)). The similarity of amino acid sequences was determined by using DNAMAN v6 software for multiple sequence alignments. MEGA 4.0 software was used for generating the phylogenetic tree.

### 4.2. Stress Treatment of B73

The maize inbred line B73 was used for investigating the expression pattern of *ZmGLYI-8* in different tissues and under various stress treatments. The seeds of inbred line B73 were preserved in our laboratory. Roots, stems and leaves of B73 at the V6 stage were collected for the expression pattern analysis of different tissues. B73 seeds were germinated in quartz sand, and the plants were cultivated at 25 °C under a 14 h light/10 h dark cycle. After ten days, the seedlings were transferred to Hoagland solution supplemented with 150 mM NaCl, 10% (*w*/*v*) PEG-6000, 10 mM MG, 0.5 mM H_2_O_2_, 50 µM ABA, 50 µM MeJA or 5 mM ET. Other seedlings were subjected to 38 °C for the high-temperature (HT) treatment. Leaf samples were collected 0, 1, 6, 12 and 24 h after the different treatments. Each biological replicate contained three plants, and all *ZmGLYI-8* expression pattern experiments were performed by using three biological replicates. All samples were frozen directly in liquid nitrogen and then stored at −80 °C before RNA extraction. The specific gene primers used for expression pattern analysis are listed in Appendix A.

### 4.3. Subcellular Localization

The subcellular localization of *ZmGLYI-8* was predicted by the SubLoc program. Moreover, the expression vector PROKII-GFP was used to investigate the subcellular localization of *ZmGLYI-8*. The coding sequence (CDS) of *ZmGLYI-8* without a termination codon was amplified by polymerase chain reaction (PCR) and then fused to the PROKII-GFP vector and driven by the CaMV35S promoter. The *ZmGLYI-8* primers used for plasmid construction are listed in Appendix A. The *Agrobacterium tumefaciens* strains GV3101 with the 35S::*ZmGLYI-8*-GFP plasmid or 35S::GFP plasmid were injected into four-leaf-stage *Nicotiana benthamiana* plants [65]. Three days after infiltration, GFP fluorescence was observed by using an LSM 880 confocal laser scanning microscope.

### 4.4. Prokaryotic Expression of ZmGLYI-8

The full-length CDS of *ZmGLYI-8* was fused to the pET-30a vector and then transformed into *Escherichia coli* Rosetta (Appendix A). PCR identification and sequencing analysis (Boshang) were used for selecting positive clones. The IPTG-induced bacteria were diluted with LB liquid medium in gradients of 10^−2^, 10^−4^ and 10^−6^. Three microliters of different concentrations of bacteria was added dropwise to LB solid medium supplemented with 0.5 mM MG, 200 mM NaCl or 200 mM mannitol. The growth of the recombinant strain pET-30a-*ZmGLYI-8* and the control strain pET-30a was observed after culture at 37 °C for 12 h.

### 4.5. Development of Transgenic Arabidopsis

The coding sequence of the *ZmGLYI-8* gene was cloned and inserted into pBI121 with the CaMV35S promoter. The resulting plasmid was subsequently introduced into *Agrobacterium tumefaciens* GV3101, which was subsequently used for floral dip transformation of *Arabidopsis* Columbia-0 (WT) plants. Transgenic *Arabidopsis* plants were screened on 1/2 MS solid medium supplemented with 50 mg/L kanamycin. The expression level of *ZmGLYI-8* in positive transgenic plants was further measured by qRT–PCR. The primers used are shown in Appendix A. T_3_ transgenic *Arabidopsis* plants were used for subsequent experimentation. Fifty seeds of WT and *ZmGLYI-8* transgenic *Arabidopsis* plants were sown on 1/2 MS solid medium supplemented with 5 mM MG. These petri dishes were placed in a growth chamber at 22 °C with a 16-h light/8-h dark cycle. The germination rate of the seeds was calculated every day, and photos were taken on the 7th day after sowing.

### 4.6. Expression Analysis by qRT–PCR

Total RNA from maize and *Arabidopsis* plants was extracted for qRT–PCR using the TRIzol reagent (TaKaRa, Beijing, China). HiScript^®^II Q Select RT SuperMix (Vazyme, China) was used for synthesizing cDNA from the total RNA. According to the manufacturers’ protocols, qRT–PCR experiments were performed by using a Bio-Rad CFX96 real-time system with AceQ^®^qPCR SYBR Green Master Mix (Vazyme, China). The inner reference gene *Actin* was used for normalizing the expression levels of genes. The gene expression levels were calculated by using the 2^−ΔΔCt^ method [66]. Triplicate independent biological assays of each cDNA sample were performed to ensure accurate statistical analysis.

### 4.7. Assays for Salt and Drought Tolerance

To detect seed germination of *ZmGLY1-8* transgenic *Arabidopsis* lines under MG stress, the sterile seeds of the WT and transgenic *Arabidopsis* lines (OE3, OE4 and OE5) were sown on 1/2 MS solid medium with MG (0 and 2.5 mM). The number of seed germinations was recorded every day, and the photos were taken on the 7th day after the seeds were sown. For the root length experiment, the sterilized WT and transgenic line seeds were sown on 1/2 MS medium for vernalization at 4 °C for 3 days. Then, these seeds were vertically cultivated at 22 °C under a 16 h light/8 h dark cycle. After five days, the seedlings were transplanted into 1/2 MS solid medium supplemented with 0.5 mM MG, 150 mM NaCl and 200 mM mannitol [39,67]. The primary root lengths of the transgenic *Arabidopsis* lines were analyzed by using ImageJ 1.50I software after 7 days of stress treatment. The experiment was repeated with three biological replicates.

### 4.8. Floating Leaf Disc and Chlorophyll Extraction

Leaf discs (1 cm in diameter) were excised from the third and fourth true rosette leaves of the WT and *ZmGLY1-8* transgenic plants. The discs were placed in water (simulated control) or under multiple stress conditions (simulated stress treatment) for 48 h, including salt stress in 200 mM and 400 mM NaCl, drought stress in 200 mM and 400 mM mannitol and MG stress in 1 mM and 2.5 mM. The chlorophyll content was determined by an ultraviolet spectrophotometer after extraction of all the leaf discs with 95% ethanol [68]. The experiment was repeated with three biological replicates.

### 4.9. Physiological Analysis and MG Content Determination

Four-week-old *Arabidopsis* plants were treated with 400 mM NaCl or 200 mM 10% PEG-6000 for 7 days, after which leaves from the same parts were removed for physiological measurements. The contents of O_2_^•−^ and H_2_O_2_ were measured using kits purchased from Suzhou Grace Biotechnology Co., Ltd., Suzhou, China. About 0.1g of the sample was taken, the extract was added, and the supernatant was collected as the solution to be measured after centrifugation. The reaction solution was added according to the kit instructions, and O_2_^•−^ and H_2_O_2_ were measured at 540 nm and 415 nm, respectively. The MDA content was also measured using kits from the same company. MDA was measured according to the kit instructions. First, 0.1 g of sample was taken, 1 mL of extraction solution was added, and the supernatant was collected after centrifuged at 12,000× *g* for 10 min at 4 °C. Next, 600 mL of working solution was added, and the supernatant was removed from the water bath at 90 °C for 30 min. MDA was measured at 532 nm and 600 nm. The experiment was conducted using three biological replicates. The activities of CAT, POD and SOD were determined by Wu’s method [69]. In order to determine the CAT, POD and SOD, fresh samples were taken, and 5 mL pre-cooled phosphoric acid buffer was added. Then, the mixture was ground mechanically and centrifuged at 10,000× *g* for 20 min at 4 °C. The collected supernatant was used for subsequent assays. SOD was measured at 560 nm using the nitrogen blue tetrazolium (NBT) method. POD and CAT activities were determined at 470 nm and 240 nm, respectively, with absorbance changes every 10 s. The methylglyoxal content was determined according to previously described methods [11]. MG content was measured from 300 mg of fresh leaf tissue. First, 0.5 M of 3 mL perchloric acid was added, ice-bathed for 15 min and centrifuged at 12,000× *g* for 10 min at 4 °C. Second, the colored supernatant was transferred to a new centrifuge tube, bleached with charcoal (10 mg/mL) for 15 min at room temperature and centrifuged at 11,000× *g* for 10 min. Then, the supernatant was transferred to a new centrifuge tube, and the reaction was neutralized by the addition of saturated potassium carbonate at room temperature for 15 min and centrifuged at 11,000× *g* for 10 min; 650 μL of supernatant was collected. Finally, the reaction mixture (1 mL), containing 250 mL of 7.2 mM 1,2-diaminobenzene, 100 mL of 5 M perchloric acid, 10 mL of 100 mM NaN_3_ and 650 μL of neutralized supernatant, was incubated for 3 h at room temperature, and the absorbance was measured at 336 nm.

### 4.10. Statistical Analysis

All data are presented as the mean of three biological replicates ± SD. One-way ANOVA and Student’s *t*-test were used for statistical analysis, and *p* < 0.05 is considered significant [70]. One-way ANOVA was performed using SPSS 22.0, and the other analyses were performed using Microsoft Office Excel 2010.

## Figures and Tables

**Figure 1 ijms-25-10937-f001:**
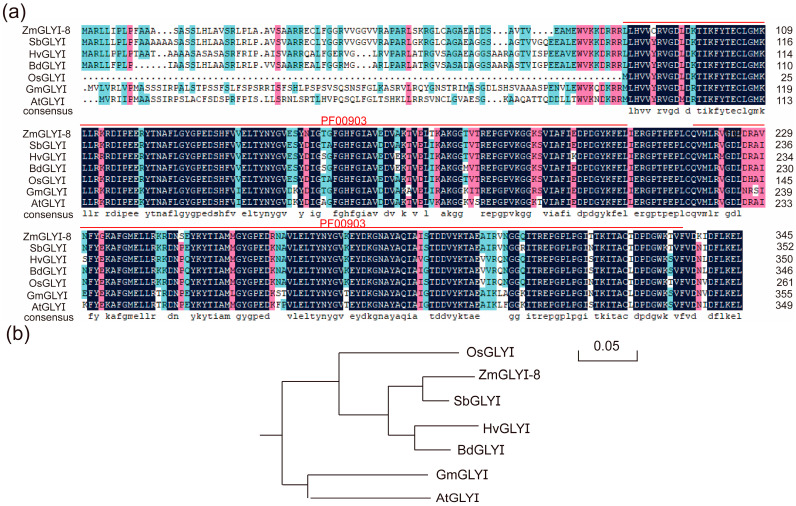
Amino acid sequence alignment and phylogenetic analysis of *ZmGLYI-8*. (**a**) Alignment of the amino acid sequence of *ZmGLYI-8* with those of its homologous proteins. (**b**) Phylogenetic tree of *ZmGLYI-8* in different plant species. The accession numbers of the selected ZmGLYIs were as follows: *AtGLYI* (NP-176896.1), *Arabidopsis*; *GmGLYI* (XP-003530499), *Glycine max*; *OsGLYI* (LOC4338161), *Oryza sativa*; *BdGLYI* (XM-003508671.1), *Brachypodium distachyon*; *HvGLYI* (AK364964.1), *Hordeum vulgare* and *SbGLYI* (XM002440785.1), *Sorghum bicolor*.

**Figure 2 ijms-25-10937-f002:**
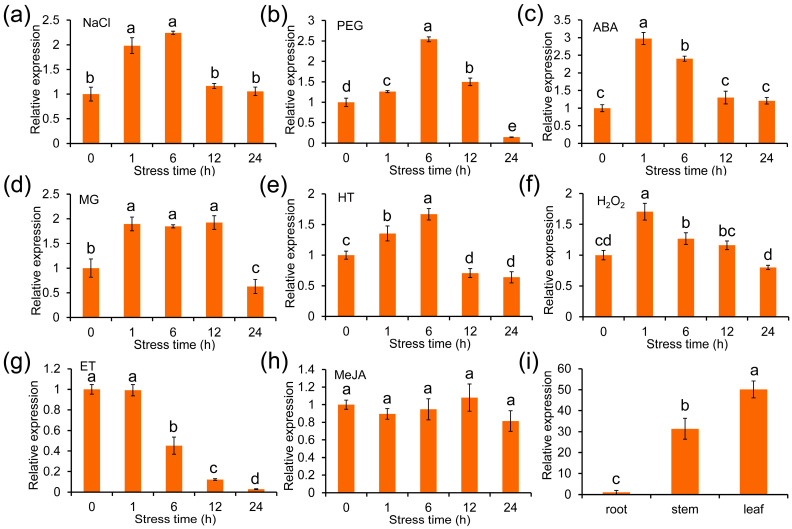
Analysis of *ZmGLYI-8* expression in maize seeds. (**a**–**h**) Expression patterns of *ZmGLYI-8* under 150 mM NaCl (**a**), 10% PEG-6000 (**b**), 50 µM ABA (**c**), 10 mM MG (**d**), HT (38 °C) (**e**), 0.5 mM H_2_O_2_ (**f**), 5 mM ET (**g**) and 50 µM MeJA (**h**) treatment. (**i**) Expression levels of *ZmGLYI-8* in roots, stems and leaves. Different letters above columns indicate significance difference (*p* < 0.05).

**Figure 3 ijms-25-10937-f003:**
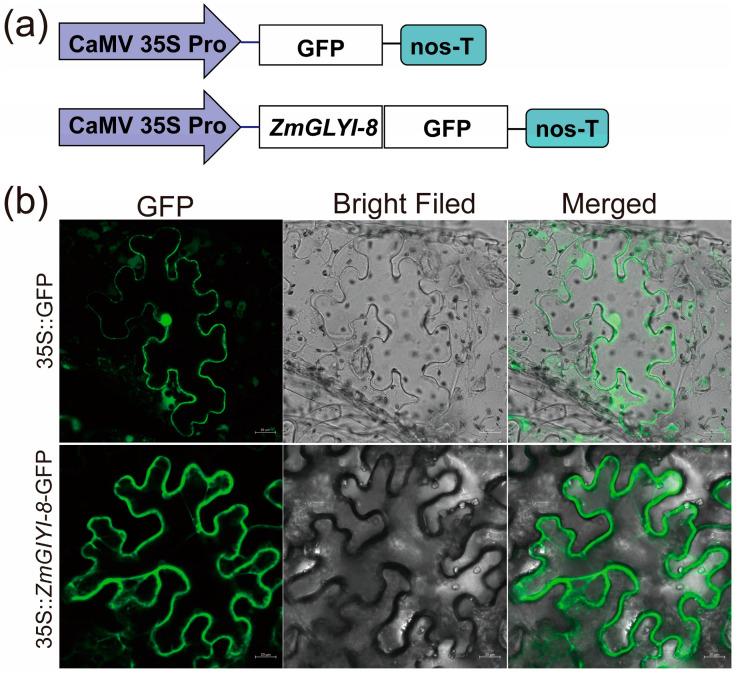
(**a**) Cytoplasmic localization of ZmGLYI-8. (**b**) The construct 35S:GFP: ZmGLYI-8 and the control vector 35S:GFP were transformed into *Nicotiana tabacum* leaves. Scale bar: 20 µm.

**Figure 4 ijms-25-10937-f004:**
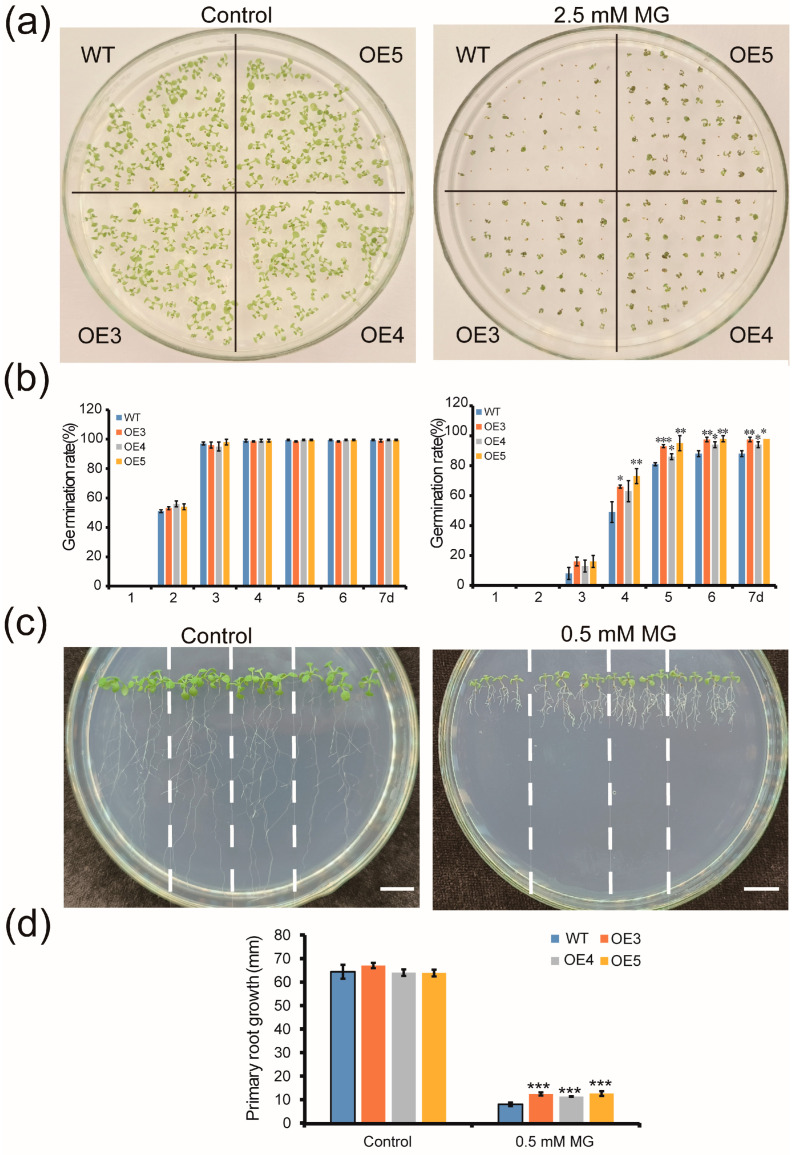
Phenotypic analysis of WT and transgenic *Arabidopsis* plants under MG treatment. (**a**) Seed germination of the WT, OE3, OE4 and OE5 plants treated with or without 2.5 mM MG for 7 days. (**b**) Seed germination rates in WT and transgenic seeds. (**c**) WT, OE3, OE4 and OE5 seedlings grown with or without 0.5 mM MG for 10 days. (**d**) Lengths of the primary roots of the WT and OE3, OE4 and OE5 plants WT, wild type. * *p* < 0.05, ** *p* < 0.01, *** *p* < 0.001. Scale bar: 10 mm.

**Figure 5 ijms-25-10937-f005:**
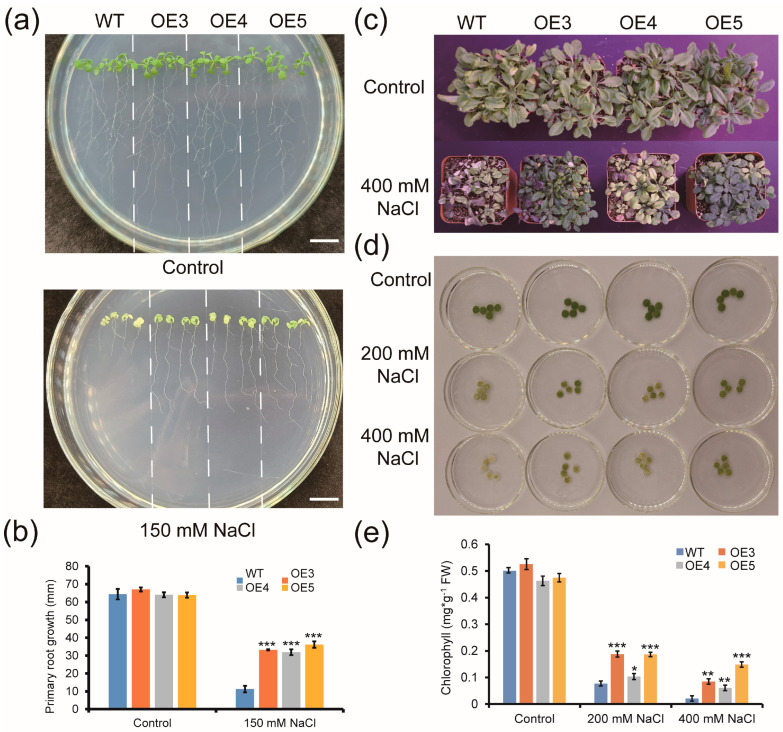
Phenotypic analysis of WT and transgenic *Arabidopsis* plants under salt treatment. (**a**,**b**) Phenotype and root length statistics of WT, OE3, OE4 and OE5 seedlings grown with or without 150 mM NaCl for 10 days. (**c**) Phenotypes of WT, OE3, OE4 and OE5 plants treated with or without 400 mM NaCl for 2 weeks. (**d**,**e**) Leaf disc assay and chlorophyll content under control and stress conditions (200 and 400 mM NaCl). * *p* < 0.05, ** *p* < 0.01, *** *p* < 0.001. Scale bar: 10 mm.

**Figure 6 ijms-25-10937-f006:**
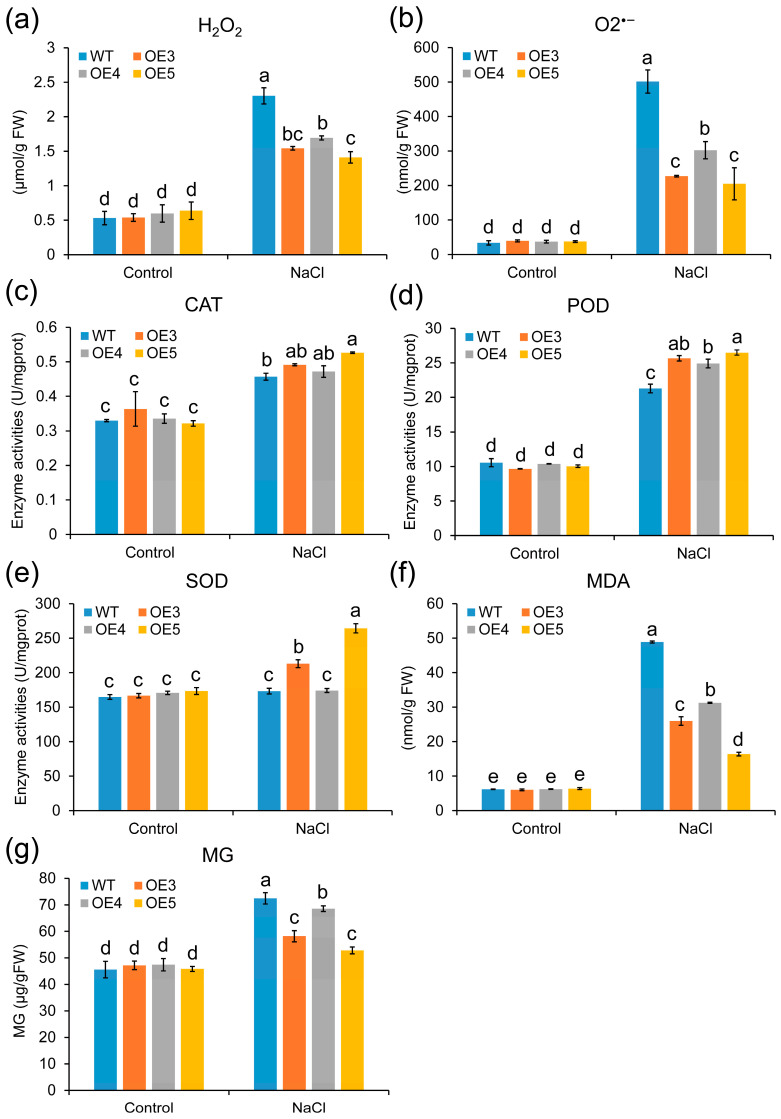
Physiological parameters after 7 days of salt treatments. (**a**,**b**,**f**) Oxidative stress indicators H_2_O_2_, O_2_^•−^ and MDA content, respectively. (**c**–**e**) CAT, POD and SOD antioxidative enzyme activities, respectively. (**g**) MG content. Different letters above columns indicate significance difference (*p* < 0.05).

**Figure 7 ijms-25-10937-f007:**
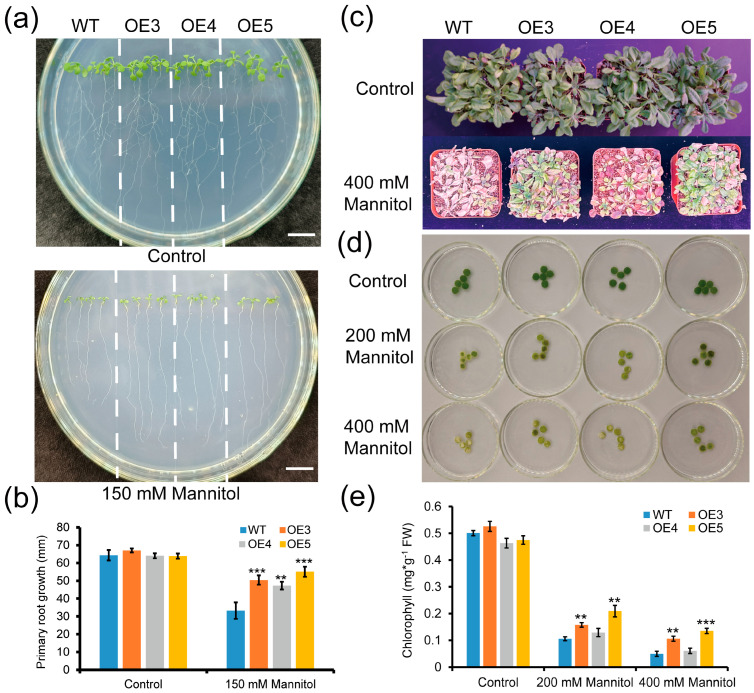
Phenotypic analysis of WT and transgenic *Arabidopsis* plants under drought treatment. (**a**,**b**) Phenotype and root length statistics of WT, OE3, OE4 and OE5 seedlings grown with or without 150 mM mannitol for 10 days. (**c**) Phenotypes of WT, OE3, OE4 and OE5 plants treated with or without 400 mM mannitol for 2 weeks. (**d**,**e**) Leaf disc assay and chlorophyll content under control and stress conditions (200 and 400 mM mannitol). ** *p* < 0.01, *** *p* < 0.001. Scale bar: 10 mm.

**Figure 8 ijms-25-10937-f008:**
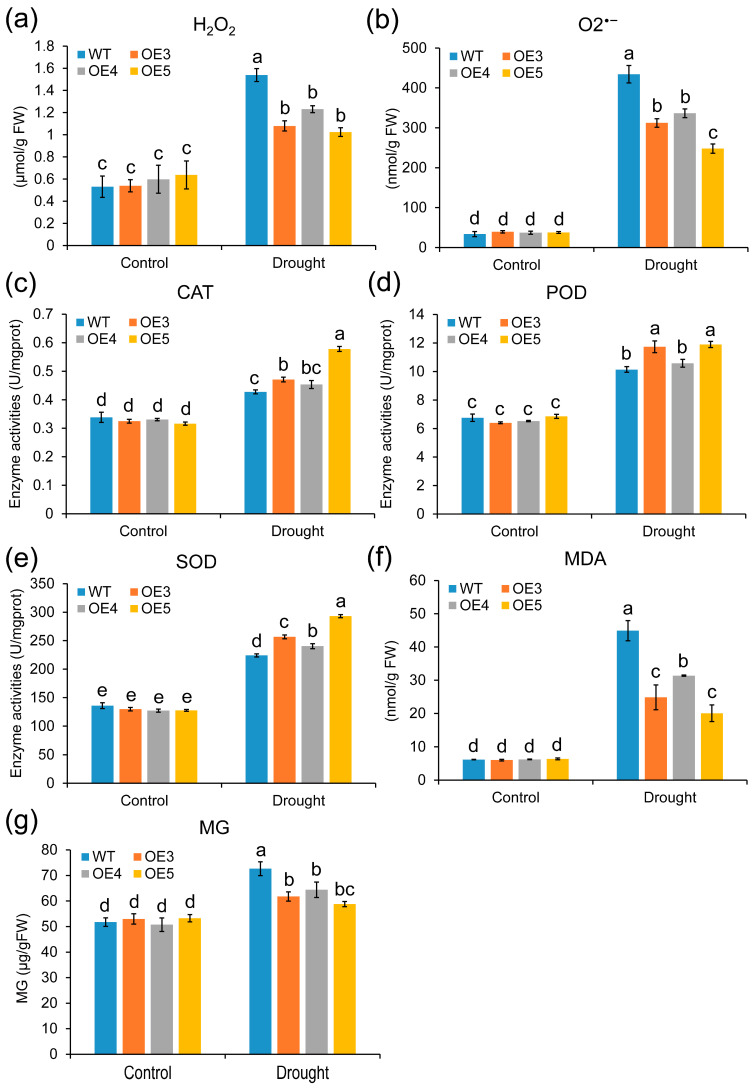
Physiological parameters after 7 days of drought treatments. (**a**,**b**,**f**) Oxidative stress indicators H_2_O_2_, O_2_^•−^ and MDA content, respectively. (**c**–**e**) CAT, POD and SOD antioxidative enzyme activities, respectively. (**g**) MG content. Different letters above columns indicate significance difference (*p* < 0.05).

**Figure 9 ijms-25-10937-f009:**
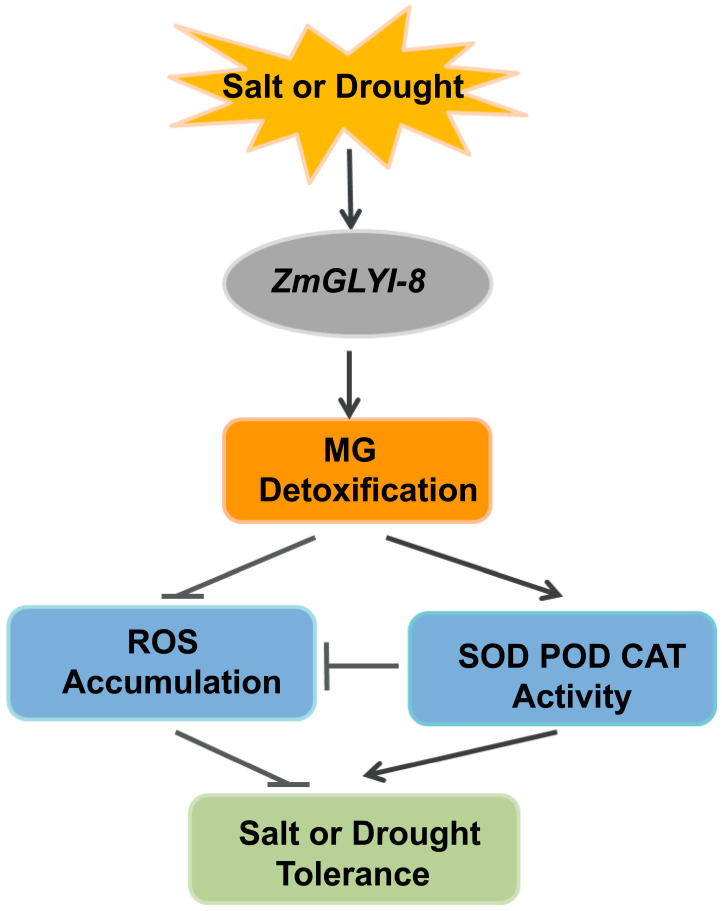
The molecular mechanism model for the regulation of plant salt and drought stress by *ZmGLYI-8*. When plants were subjected to salt and drought stress, overexpression of the *ZmGLYI-8* gene was able to detoxify the overproduced MG in transgenic *Arabidopsis*, which in turn increased the activities of SOD, POD and CAT, reduced the accumulation of ROS and ultimately improved the tolerance of transgenic *Arabidopsis* to salt and drought stress.

## Data Availability

Data is contained within the article or Appendix A.

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
