# Peer review of "The Maize Gene ZmGLYI-8 Confers Salt and Drought Tolerance in Transgenic Arabidopsis Plants"

_ijms, 2024, doi:10.3390/ijms252010937_

Round 1
Reviewer 1 Report
Comments and Suggestions for Authors
The authors demonstrated that overexpression of maize ZmGLYI-8 enhances salt and drought tolerance in transgenic Arabidopsis plants. I recommend the manuscript with minor suggestions.
Discuss in more detail how overexpression of ZmGLYI-8 influences reactive oxygen species (ROS) levels and antioxidant enzyme activity in transgenic Arabidopsis plants.
What molecular mechanisms were elucidated in this study regarding the maize ZmGLYI-8 gene? I suggest including a figure to illustrate these mechanisms.
Improve the discussion by incorporating the latest references. Rewrite the conclusion with information related how this study significantly advance our knowledge and contribute to imrpove the maize abiotic stress tolerance.
Author Response
Please see the author response in the attachment.

Reviewer 2 Report
Comments and Suggestions for Authors
Dear Authors,
I have an opportunity to review manuscript entitled:” The maize gene ZmGLYІ-8 confers salt and drought tolerance in transgenic Arabidopsis plants” submitted to IJMS.
Authors focused on Glyoxalase І (GLYІ), the first enzyme of the glyoxalase pathway, plays multiple roles in the detoxification of MG and in abiotic stress responses. Moreover, screened a glyoxalase І gene (ZmGLYІ-8), overexpressed in Arabidopsis and test it under multiple: salt, drought, and many more treatments.
Introduction part seems to be clear and efficient to analyses Author’s obtained results.
· I understand that “few studies have confirmed the ability of multiple glyoxalase genes to enhance abiotic stress resistance in plants, the underlying genetic mechanism is unclear. Whether the remaining 13 GLYI genes and 3 GLYII genes in maize respond to abiotic stress has not been reported”, but the reader should have an explanation why exactly Authors chose GLYI-8 gene??
· Please, underline the precise aim of the studies in presented work, not only in abstract part;
Materials and methods could be written more precisely, especially 2.9 part
· Statements „The contents of O•−2 and H2O2 were measured using kits purchased from Grace Biotechnology. The MDA content was also measured using kits from the same company.” Are inefficient; Please, add methodology or cite protocols based on previous literature.
I have some doubts to Figure 2 and 6 with 8 regarding statistical analyses, Please, use appropriate test to have reliability data about differences in statistical significance. Please, do not only use standard deviations especially in relative genes expression.
Please, analyse carefully the GFP-localisation and be precise, is it really localization in cytoplasm ?
The second question is what kind of reference genes were used, because in supplementary files I find “action F/ action R” ??
The discussion part is quite short and concrete; I'm curious what Authors think, what kind of future prospects came form Authors new findings to make their results even more interesting to the wider audience.
Author Response
- Discuss in more detail how overexpression of ZmGLYI-8 influences reactive oxygen species (ROS) levels and antioxidant enzyme activity in transgenic Arabidopsis plants.
RESPONSE: Thanks for your nice suggestion. We have added overexpression of ZmGLYI-8 influences reactive oxygen species (ROS) levels and antioxidant enzyme activity in transgenic Arabidopsis plants in lines 465-513.
- What molecular mechanisms were elucidated in this study regarding the maize ZmGLYI-8 gene? I suggest including a figure to illustrate these mechanisms.
RESPONSE: Following your suggestion we added a molecular mechanism model diagram, Figure 9, in the discussion part to elucidate the molecular mechanisms of the ZmGLYI-8 gene (Lines 534-540).
- Improve the discussion by incorporating the latest references. Rewrite the conclusion with information related how this study significantly advance our knowledge and contribute to imrpove the maize abiotic stress tolerance.
RESPONSE: Thanks for your nice suggestion. We have added some recent references to the discussion part, such as references 68-71. Also some of the conclusions in the discussion part have been rewritten in lines 491-493, 510-513 and 528-533, In addition, We have rewrite the conclusion with information related how this study significantly advance our knowledge and contribute to imrpove the maize abiotic stress tolerance in lines 517-533.
Round 2
Reviewer 2 Report
Comments and Suggestions for Authors
Dear Authors,
I'm really sorry, but Author's give response not to my revision; Moreover, in manuscript there are lack of tracking changes.
Therefore, I am not able to decide about manuscript corrections and I'm not changing my decision
Author Response
- I understand that “few studies have confirmed the ability of multiple glyoxalase genes to enhance abiotic stress resistance in plants, the underlying genetic mechanism is unclear. Whether the remaining 13 GLYI genes and 3 GLYII genes in maize respond to abiotic stress has not been reported”, but the reader should have an explanation why exactly Authors chose GLYI-8 gene?
RESPONSE: Thanks for your comment. In the preliminary stage of this study, transcriptome sequencing was performed on maize seedlings after MG treatment, and glyoxalase family genes were analyzed, and it was that ZmGLYІ-8 gene induced expression after MG treatment. At present, the transcriptome data is unpublished.
- Please, underline the precise aim of the studies in presented work, not only in abstract part.
RESPONSE: We have added the precise aim of the studies in presented work in lines 87-98.
- Materials and methods could be written more precisely, especially 2.9 part.
RESPONSE: We have added the precise material method in Part 2.9 (Lines 195-222).
- Statements, The contents of O•−2 and H2O2 were measured using kits purchased from Grace Biotechnology. The MDA content was also measured using kits from the same company.” Are inefficient; Please, add methodology or cite protocols based on previous literature.
RESPONSE: Thanks for your nice suggestions. We have added specific methods for the determination of O•−2, H2O2 and MDA in Part 2.9 (Lines 195-203).
- I have some doubts to Figure 2 and 6 with 8 regarding statistical analyses, Please, use appropriate test to have reliability data about differences in statistical significance. Please, do not only use standard deviations especially in relative genes expression.
RESPONSE: Thanks for your suggestion. We have reanalyzed the differences in statistical significance in Figures 2, 6 and 8 using one-way ANOVA method. And we have replaced the images of Figures 2, 6 and 8 in the manuscript.
- Please, analyse carefully the GFP-localisation and be precise, is it really localization in cytoplasm?
RESPONSE: Yes, the ZmGLYІ-8 protein is localized in the cytoplasm. Prokaryotic and human GLYI proteins are known to be present in the cytosol of the cells as their substrate MG is primarily produced in a glycolytic bypass. Similarly, most GLY proteins in plants are found in the cytoplasm. From Figure 3B, we can see that ZmGLYI-8 protein has scattered fluorescence presence in the cytoplasm. Moreover, we analyzed the localization of ZmGLYI-8 protein using SubLoc online prediction software and found that the protein was localized in the cytoplasm.
- The second question is what kind of reference genes were used, because in supplementary files I find “action F/ action R”?
RESPONSE: We are sorry for our carelessness. “action F/ action R” should be “actin F/actin R” for primers specific to inner reference gene β-Actin. We have made corrections in the supplementary files.
- The discussion part is quite short and concrete; I'm curious what Authors think, what kind of future prospects came form Authors new findings to make their results even more interesting to the wider audience.
RESPONSE: Thank you for your nice suggestions. We have enriched the discussion part and added the future prospects offered in the last paragraph of the discussion to make our results more interesting to the wider audience (Lines 517-533).